# Development of a Multiplex Crystal Digital RT-PCR for Differential Detection of Classical, Highly Pathogenic, and NADC30-like Porcine Reproductive and Respiratory Syndrome Virus

**DOI:** 10.3390/ani13040594

**Published:** 2023-02-08

**Authors:** Feng Long, Yating Chen, Kaichuang Shi, Yanwen Yin, Shuping Feng, Hongbin Si

**Affiliations:** 1Guangxi Center for Animal Disease Control and Prevention, Nanning 530001, China; 2College of Animal Science and Technology, Guangxi University, Nanning 530005, China

**Keywords:** porcine reproductive and respiratory syndrome virus (PRRSV), multiplex real-time quantitative RT-PCR (multiplex qRT-PCR), multiplex Crystal digital RT-PCR (multiplex cdRT-PCR), classical PRRSV (C-PRRSV), highly pathogenic PRRSV (HP-PRRSV), NADC30-like PRRSV (NL-PRRSV), differential detection

## Abstract

**Simple Summary:**

Porcine reproductive and respiratory syndrome (PRRS) is one of the most important diseases of swine in the world. PRRS virus (PRRSV) type 1 (European genotype, EU-PRRSV) and PRRSV type 2 (North American genotype, NA-PRRSV) are simultaneously prevalent in China nowadays. Of the NA-PRRSV, classical PRRSV (C-PRRSV), highly pathogenic PRRSV (HP-PRRSV), and NADC30-like PRRSV (NL-PRRSV) are the most common circulating strains in China. Here, a multiplex real-time quantitative RT-PCR (qRT-PCR) and a multiplex Crystal digital RT-PCR (cdRT-PCR) were developed for the differential detection of C-PRRSV, HP-PRRSV, and NL-PRRSV. A total of 320 clinical samples were used to evaluate the application of these developed assays, and the positive rates of C-PRRSV, HP-PRRSV, and NL-PRRSV by the multiplex qRT-PCR were 1.88%, 21.56%, and 9.69%, respectively, while the positive rates by the multiplex cdRT-PCR were 2.19%, 25.31%, and 11.56%, respectively. These two assays showed high sensitivity, strong specificity, and excellent repeatability for the simultaneous and differential detection of C-PRRSV, HP-PRRSV, and NL-PRRSV.

**Abstract:**

Porcine reproductive and respiratory syndrome virus (PRRSV) type 1 (European genotype) and PRRSV type 2 (North American genotype) are prevalent all over the world. Nowadays, the North American genotype PRRSV (NA-PRRSV) has been widely circulating in China and has caused huge economic losses to the pig industry. In recent years, classical PRRSV (C-PRRSV), highly pathogenic PRRSV (HP-PRRSV), and NADC30-like PRRSV (NL-PRRSV) have been the most common circulating strains in China. In order to accurately differentiate the circulating strains of NA-PRRSV, three pairs of specific primers and corresponding probes were designed for the Nsp2 region of C-PRRSV, HP-PRRSV, and NL-PRRSV. After optimizing the annealing temperature, primer concentration, and probe concentration, a multiplex real-time quantitative RT-PCR (qRT-PCR) and a multiplex Crystal digital RT-PCR (cdRT-PCR) for the differential detection of C-PRRSV, HP-PRRSV, and NL-PRRSV were developed. The results showed that the two assays illustrated high sensitivity, with a limit of detection (LOD) of 3.20 × 10^0^ copies/μL for the multiplex qRT-PCR and 3.20 × 10^−1^ copies/μL for the multiplex cdRT-PCR. Both assays specifically detected the targeted viruses, without cross-reaction with other swine viruses, and indicated excellent repeatability, with coefficients of variation (CVs) of less than 1.26% for the multiplex qRT-PCR and 2.68% for the multiplex cdRT-PCR. Then, a total of 320 clinical samples were used to evaluate the application of these assays, and the positive rates of C-PRRSV, HP-PRRSV, and NL-PRRSV by the multiplex qRT-PCR were 1.88%, 21.56%, and 9.69%, respectively, while the positive rates by the multiplex cdRT-PCR were 2.19%, 25.31%, and 11.56%, respectively. The high sensitivity, strong specificity, excellent repeatability, and reliability of these assays indicate that they could provide useful tools for the simultaneous and differential detection of the circulating strains of C-PRRSV, HP-PRRSV, and NL-PRRSV in the field.

## 1. Introduction

Porcine reproductive and respiratory syndrome (PRRS) causes reproductive disorders in sows and respiratory diseases and a reduction in growth performances in pigs of all ages, which has caused huge economic losses to the pig industry worldwide [1]. PRRS virus (PRRSV) is an enveloped RNA virus with a single-strand, positive-sense genome of about 15 kb, and is a member of the order *Nidovirales*, family *Arteriviridae*, and genus *Arterivirus* [2]. On the basis of genetic and antigenic differences, PRRSV is divided into two types: genotype 1 (European genotype, EU-PRRSV), which is represented by the Lelystad virus (LV), and genotype 2 (North American genotype, NA-PRRSV), which is represented by the VR-2332 strain [3]. PRRSV first emerged in North America in 1987 [4] and was first reported in China in 1995 [5]. Highly pathogenic PRRSV (HP-PRRSV), which is characterized by two discontinuous deletions of 30 amino acids (29 aa plus 1 aa) in the nonstructural protein 2 (NSP2) compared with classical PRRSV (C-PRRSV), was first identified in 2006 in China [6]. The NADC30 strain was first discovered in the United States of America (USA) in 2008 [7]. In 2013, the NADC30-like PRRSV (NL-PRRSV), which is characterized by a unique three discontinuous deletions of 131 amino acids (111 aa plus 1 aa plus 19 aa) in the NSP2, was first identified in China and caused respiratory signs with a mortality rate of up to 30–50% [8]. Nowadays, EU-PRRSV and NA-PRRSV have been simultaneously circulating in China, and most cases of PRRS have been caused by NA-PRRSV [9,10]. Because of the high rates of mutations, insertions, and deletions in different viral genes and the high incidence of genetic recombination between different strains of PRRSV, circulating PRRSV in China has shown high genetic diversity, and many PRRSV strains have been identified, including Ch-1a-like, VR-2332-like, JXA1-like, NADC30-like, NADC34-like, QYYZ-like, and MLV-like PRRSV [9,10,11,12]. Of which, the C-PRRSV (VR-2332-like), HP-PRRSV (JXA1-like), and NL-PRRSV (NADC30-like) strains of NA-PRRSV have been the most common strains currently circulating in China [9,10,13,14,15,16], and they have caused similar signs and damage to the infected pigs. It is necessary to develop a specific, accurate, and sensitive assay for the differential detection of C-PRRSV, HP-PRRSV, and NL-PRRSV.

Real-time quantitative PCR/RT-PCR (qPCR/qRT-PCR) has the advantages of a low chance of contamination, fast reaction speed, and high sensitivity and has been widely used for the detection of viral nucleic acids [17]. The newly emerging digital PCR/RT-PCR (dPCR/dRT-PCR), the third generation PCR/RT-PCR developed on the basis of qPCR/qRT-PCR, uses a droplet generator to decompose the reaction system, and DNA/RNA is encapsulated stochastically inside the microdroplets as reaction chambers, with a small percentage of the reaction chamber containing one or fewer copies of the DNA or RNA [18,19]. After amplification, the concentrations of the targeted gene are determined based on the proportion of nonfluorescent partitions through Poisson distribution using a droplet analyzer to detect the fluorescence signal of each droplet. Compared with qPCR/qRT-PCR, dPCR/dRT-PCR is an endpoint measurement that is capable of absolute quantification independent of reference genes, standard curves, and Ct values with high sensitivity, specificity, precision, efficacy, and good tolerance to inhibitors [19,20]. Recently, dPCR/dRT-PCR has been gradually used in many laboratories [21,22]. To date, multiplex qRT-PCR has been developed for the detection of different genotypes of PRRSV [23,24,25,26] and different strains of NA-PRRSV [26,27,28,29,30]. dRT-PCR has been developed for the detection of PRRSV [31], and multiplex Crystal dPCR (cdPCR) has been developed for the detection of African swine fever virus (ASFV), classical swine fever virus (CSFV), and PRRSV [32]. However, no multiplex cdRT-PCR for the detection of different NA-PRRSV strains has been reported until now. The Naica System^TM^ for Crystal Digital^TM^ PCR performs digital PCR using a hybrid approach, named Crystal Digital^TM^ PCR, combining the 2D array format of chamber dPCR and the use of droplet partitions, as implemented in droplet dPCR [33]. Crystal Digital^TM^ PCR, which relies on the use of a single chip to partition samples into 2D droplet arrays, is then subjected to thermal cycling and, finally, read using a three-color fluorescence scanning device. This novel technology, thus, allows three-color multiplexing, which entails a different approach to data analysis [34]. In this study, a multiplex qRT-PCR and a multiplex cdRT-PCR were developed for the differential detection of C-PRRSV, HP-PRRSV, and NL-PRRSV and used to test 320 clinical samples from Guangxi province, Southern China.

## 2. Materials and Methods

### 2.1. Viral Strains

The following vaccine strains were purchased from Huapai Bioengineering Group Co. Ltd. (Chengdu, China) and other companies: Ch-1R strain of C-PRRSV, TJM92 strain of HP-PRRSV, C strain of CSFV, O/Mya98/XJ/2010 strain of foot-and-mouth disease virus (FMDV), ZJ/C strain of porcine circovirus type 2 (PCV2), Bartha-K61 strain of porcine pseudorabies virus (PRV), SC1 strain of porcine parvovirus (PPV), and TJ strain of swine influenza virus (SIV). The positive clinical samples of NL-PRRSV, ASFV, and Senecavirus A (SVA) were obtained by our laboratory and confirmed by genomic sequencing. They were stored at −70 °C until used.

### 2.2. Clinical Samples

From January 2020 to December 2021, a total of 320 clinical samples (including lung, liver, spleen, kidney, tonsil, and lymph nodes of each dead pig) from 320 dead pigs were collected from 86 pig farms in Guangxi province, Southern China. The samples were transported to our laboratory under ≤4 °C and stored at −70 °C until used. Written informed consent to use the clinical samples in this study was obtained from the owners of the animals.

### 2.3. Design of Primers and Probes

The Nsp2 region was selected as the targeted region to design the specific primers and probes for differential detection of the C-PRRSV, HP-PRRSV, and NL-PRRSV strains of NA-PRRSV (Table 1). All primers and probes were synthesized by TaKaRa Biomedical Co., Ltd. (TaKaRa, Dalian, China).

### 2.4. Extraction of Nucleic Acids

All vaccine viruses and the pooled clinical tissue homogenates (20%, W/V) were resuspended in phosphate-buffered saline (PBS, pH 7.2), vortexed for 5 min, and centrifuged at 12,000× *g* at 4 °C for 10 min. The total nucleic acids were extracted from the treated samples or vaccine solution using MiniBEST Viral RNA/DNA Extraction Kit Ver.5.0 (TaKaRa, Dalian, China) and then reverse transcribed to cDNA using PrimeScript II 1st Strand cDNA Synthesis Kit (TaKaRa, Dalian, China), according to the manufacturer’s instructions. The obtained cDNA was stored at −70 °C until used.

### 2.5. Construction of the Standard Plasmids

The total nucleic acids were extracted from vaccine strains of C-PRRSV, HP-PRRSV, and the positive sample of NL-PRRSV, reverse transcribed to cDNA, and then amplified by PCR using the specific primers. The amplicons were purified and cloned into pMD18-T vector (TaKaRa, Dalian, China) and transferred into *Escherichia coli* (*E. coli*) DH5α competent cells (TaKaRa, Dalian, China). The positive clones were cultured at 37 °C for 22–24 h and extracted by MiniBEST Plasmid Extraction Kit Ver.5.0 (TaKaRa, Dalian, China) for plasmid constructs. The recombinant plasmids were named p-C-PRRSV, p-HP-PRRSV, and p-NL-PRRSV and stored at −70 °C until used as standard plasmids.

The standard plasmids were quantified by ultraviolet absorbance at 260 nm and 280 nm with a NanoDrop spectrophotometer (Thermo Fisher, Waltham, MA, USA). Their concentrations were calculated according to the following formula: plasmid copy number (copies/μL) = (6.02 × 10^23^ × plasmid concentration × 10^–9^)/(660 × plasmid length).

### 2.6. Optimization of the Reaction Conditions

The reaction conditions of the multiplex qRT-PCR, including the annealing temperatures (from 55 °C to 60 °C) and the primer and probe concentrations (from 100 nM to 600 nM), were optimized using QuantStudio™ 5 qPCR detection system (ABI, Carlsbad, CA, USA). The following basic systems were used with a total volume of 25 μL: 2× One-Step RT-PCR Buffer Ⅲ (TaKaRa, Dalian, China) 12.5 μL, Ex Taq HS (5 U/μL) (TaKaRa, Dalian, China) 0.5 μL, PrimerScript RT Enzyme Mix Ⅱ (TaKaRa, Dalian, China) 0.5 μL, the mixture of three pairs of primers and three probes with different final concentrations 0.1–0.6 μL, the mixture of three standard plasmids (with 10^7^ copies/μL of each plasmid) 0.25 μL as template, and distilled water to a final volume of 25 μL. The amplification parameters were as follows: 42 °C for 5 min; 95 °C for 10 s; 40 cycles of 95 °C for 5 s and 56 °C for 34 s. The fluorescent signals were measured at the end of each cycle of the 56 °C extension step. The final reaction conditions were optimized to obtain the maximum ΔRn and the minimal threshold cycle (Ct).

Multiplex cdRT-PCR was performed using the Naica^TM^ sapphire crystal system (Stilla Technologies^TM^, Villejuif, France). The whole process was performed inside the Sapphire chip (Stilla Technologies, Villejuif, France). The samples were pipetted into the Sapphire chip (Stilla Technologies, Villejuif, France), placed onto Naica^TM^ Geode (Stilla Technologies, Villejuif, France), and then generated droplets prior to thermal cycling. After thermocycling, the chips were transferred to Naica^TM^ Prism3 (Stilla Technologies, Villejuif, France), and the images of the blue, green, and red detection channels were acquired. Finally, the concentrations of the templates were determined using Crystal Miner software (Stilla Technologies, Villejuif, France). The following basic systems were used to determine the optimal reaction conditions of the multiplex cdRT-PCR with a total volume of 25 μL: qScript XLT One-Step RT-qPCR ToughMix (Quanta Biosciences, Gaithersburg, MD, USA) 12.5 μL, Fluorescein sodium salt (1 μM) (Apexbio Biotechnology, Beijing, China) 2.5 μL, the mixture of three pairs of primers and three probes of different final concentrations 0.2–0.6 μL, the mixture of three standard plasmids (with 10^3^ copies/μL of each plasmid) 0.25 μL as template, and distilled water to a final volume of 25 μL. The amplification parameters were as follows: 42 °C for 5 min; 95 °C for 5 min; 45 cycles of 95 °C for 5 s; 60 °C for 30 s; 72 °C for 30 s.

### 2.7. Specificity Analysis

The mixture of three standard plasmids was used as positive control and sterilized distilled water was used as negative control. The DNA or cDNA of C-PRRSV, HP-PRRSV, NL-PRRSV, ASFV, PRV, CSFV, FMDV, PPV, SIV, PCV2, and SVA was used as templates to evaluate the specificity of the multiplex qRT-PCR and the multiplex cdRT-PCR.

### 2.8. Sensitivity Analysis

Three standard plasmids of p-C-PRRSV, p-HP-PRRSV, and p-NL-PRRSV were mixed together with a ratio of 1:1:1, serially diluted (10-fold), and used as templates to evaluate the sensitivity of the multiplex qRT-PCR and the multiplex cdRT-PCR.

### 2.9. Repeatability Analysis

Three standard plasmids of p-C-PRRSV, p-HP-PRRSV, and p-NL-PRRSV were mixed together and 10-fold serially diluted. Three concentrations of 3.20 × 10^3^, 3.20 × 10^2^, and 3.20 × 10^1^ copies/μL (final concentrations) were used as templates to evaluate the repeatability and reproducibility of the multiplex qRT-PCR and the multiplex cdRT-PCR. The intra-assays were run in triplicates, and the inter-assays were run on three different days.

### 2.10. Detection of the Clinical Samples

A total of 320 clinical samples, which were collected from Guangxi province, Southern China between January 2020 and December 2021, were detected by the multiplex qRT-PCR and the multiplex cdRT-PCR. The coincidence rates and Kappa values of the detection results between these two methods were calculated using SPSS version 26.0 software (IBM, Amonk, NY, USA).

## 3. Results

### 3.1. Construction of the Standard Plasmids

The total nucleic acids of C-PRRSV, HP-PRRSV, and NL-PRRSV were extracted, reverse-transcribed, and amplified by PCR using the specific primers (Table 1). The amplified fragments were purified, ligated to pMD18-T vector (TaKaRa, Dalian, China), and transferred into *E. coli* DH5α competent cells (TaKaRa, Dalian, China). The positive clones were cultured, the plasmid constructs were extracted, and their concentrations were determined by ultraviolet absorbance. The original concentrations of the standard plasmids, which were named p-C-PRRSV, p-HP-PRRSV, and p-NL-PRRSV, were 2.29 × 10^10^ copies/μL, 1.91 × 10^10^ copies/μL, and 3.12 × 10^10^ copies/μL, respectively, and then these plasmids were adjusted to the same concentration of 3.20 × 10^9^ copies/μL.

### 3.2. Determination of the Optimal Reaction Conditions

The standard plasmids were used as templates to optimize the reaction conditions of the multiplex qRT-PCR, and the optimal annealing temperature, the concentrations of primers and probes, and the cycles of amplification were obtained. The developed multiplex qRT-PCR with a total volume of 25 μL contained TaKaRa 2× One-Step RT-PCR Buffer, TaKaRa Ex Taq HS, TaKaRa PrimerScript RT Enzyme Mix (TaKaRa, Dalian, China), primers and probes for C-PRRSV, HP-PRRSV, and NL-PRRSV, total RNA as a template, and distilled water (Table 2). The amplification parameters were as follows: 42 °C for 5 min, 95 °C for 10 s, and 40 cycles of 95 °C for 5 s and 56 °C for 34 s. The sample with a Ct value ≤ 36 cycles was considered as positive, and the sample with a Ct value > 36 cycles was considered as negative.

After the optimization of different reaction conditions, the determined multiplex cdRT-PCR with a total volume of 25 μL contained qScript XLT One-Step RT-qPCR ToughMix (Quanta Biosciences, Gaithersburg, MD, USA), Fluorescein sodium salt (Apexbio Biotechnology, Beijing, China), primers and probes for C-PRRSV, HP-PRRSV, and NL-PRRSV, total RNA as a template, and distilled water. The amplification parameters were as follows: 42 °C for 5 min; 95 °C for 5 min; and 45 cycles of 95 °C for 5 s; 60 °C for 30 s; and 72 °C for 30 s. After amplification, the absolute concentration of each sample was automatically reported by the Naica^TM^ system. The positive and negative microdroplets were divided into two clusters, and the total droplets and positive droplets were generated (Figure 1).

### 3.3. Generation of the Standard Curves

The standard curves of the multiplex qRT-PCR were generated using a 10-fold serial dilution of the standard plasmids p-C-PRRSV, p-HP-PRRSV, and p-NL-PRRSV, ranging from 3.20 × 10^7^ copies/μL to 3.20 × 10^2^ copies/μL (final concentrations from 3.20 × 10^6^ copies/μL to 3.20 × 10^1^ copies/μL) (Figure 2D). The results show that the corresponding slope of the equation, the correlation coefficient (R^2^), and the amplification efficiency (E) were −3.387, 0.999, and 103.278% for C-PRRSV, −3.6, 0.999, and 107.428% for HP-PRRSV, and −3.468, 0.999, and 107.155% for NL-PRRSV, respectively (Figure 2D), indicating an excellent linear relationship between the initial concentrations of templates and the Ct values.

The standard curves of cdRT-PCR were generated using a 10-fold serial dilution of the standard plasmids p-C-PRRSV, p-HP-PRRSV, and p-NL-PRRSV, ranging from 3.20 × 10^5^ copies/μL to 3.20 × 10^0^ copies/μL (final concentrations from 3.20 × 10^4^ copies/μL to 3.20 × 10^−1^ copies/μL) (Figure 3D). The results show that the corresponding slope of the equation and the correlation coefficient (R^2^) were 1.01 and 0.9992 for C-PRRSV, 0.9732 and 0.9982 for HP-PRRSV, and 0.9926 and 0.9985 for NL-PRRSV, respectively (Figure 3D).

### 3.4. Sensitivity Analysis

The standard plasmids of p-C-PRRSV, p-HP-PRRSV, and p-NL-PRRSV were mixed together and 10-fold serially diluted from 3.20 × 10^7^ copies/μL to 3.20 × 10^0^ copies/μL (initial concentrations) and used as a template to determine the limit of detection (LOD) of the multiplex qRT-PCR. The results show that all amplification curves presented as typical S-shaped curves, and the LOD of p-C-PRRSV, p-HP-PRRSV, and p-NL-PRRSV was 3.20 × 10^0^ copies/μL (the 25 μL reaction system contained a 2.5 μL template; therefore, the final concentrations of the templates were a 10-fold dilution of the initial concentrations in the reaction system. In this study, the LOD of each template was 3.20 × 10^1^ copies/μL of the initial concentration, so the final concentration was 3.20 × 10^0^ copies/μL) (Figure 2A–C), showing the high sensitivity of the multiplex qRT-PCR.

The 10-fold serial dilution of the three standard plasmids from 3.20 × 10^6^ copies/μL to 3.20 × 10^0^ copies/μL (initial concentrations) was used as a template to determine the LOD of cdRT-PCR. The results show that the number of positive droplets decreased gradually with the decrease in concentration, and the LOD of p-C-PRRSV, p-HP-PRRSV, and p-NL-PRRSV was 3.20 × 10^−1^ copies/μL (the initial concentrations of the templates were 3.20 × 10^0^ copies/μL, so the final concentrations were 3.20 × 10^−1^ copies/μL, as abovementioned) (Figure 3A–C), indicating that the multiplex cdRT-PCR was 10 times more sensitive than the multiplex qRT-PCR.

### 3.5. Specificity Analysis

The DNA or cDNA of C-PRRSV, HP-PRRSV, NL-PRRSV, ASFV, PRV, CSFV, FMDV, PPV, SIV, PCV2, and SVA were used as templates to evaluate the specificity of the multiplex qRT-PCR and the multiplex cdRT-PCR. The results show that only C-PRRSV, HP-PRRSV, and NL-PRRSV showed typical amplification curves or positive droplets and showed no cross-reaction with other porcine viruses (Figure 4).

### 3.6. Repeatability Analysis

The concentrations of 3.20 × 10^3^, 3.20 × 10^2^, and 3.20 × 10^1^ copies/μL (final concentrations) of the standard plasmids p-C-PRRSV, p-HP-PRRSV, and p-NL-PRRSV were used to evaluate the repeatability and reproducibility of the multiplex qRT-PCR and the multiplex cdRT-PCR. The results show that the coefficients of variation (CVs) of intra-assay for repeatability and inter-assay for reproducibility were less than 1.26% and 1.70% for the multiplex qRT-PCR and less than 2.68% and 2.63% for the multiplex cdRT-PCR, respectively (Table 3).

### 3.7. Application for Detection of Clinical Samples

A total of 320 clinical samples, which were collected from January 2020 to December 2021 in Guangxi province, Southern China, were tested by the developed multiplex qRT-PCR and the multiplex cdRT-PCR. The results show that the positive rates of C-PRRSV, HP-PRRSV, and NL-PRRSV were 1.88% (6/320), 21.56% (69/320), and 9.69% (31/320) by the multiplex qRT-PCR, and 2.19% (7/320), 25.31% (81/320), and 11.56% (37/320) by the multiplex cdRT-PCR, respectively (Table 4). In addition, the positive rates of co-infections with C-PRRSV plus HP-PRRSV, C-PRRSV plus NL-PRRSV, HP-PRRSV plus NL-PRRSV, and C-PRRSV plus HP-PRRSV plus NL-PRRSV were 0.63% (2/320), 0.31% (1/320), 4.69% (15/320), and 0.31% (1/320), respectively, by the multiplex qRT-PCR, and 0.63% (2/320), 0.31% (1/320), 6.25% (20/320), and 0.31% (1/320), respectively, by the multiplex cdRT-PCR. The coincidence rates of these two methods were more than 96.25%, and the Kappa values were higher than 0.90 (Table 4). Of the 320 clinical samples, 33.13% (106/320) were positive for PRRSV by the multiplex qRT-PCR, 39.06% (125/320) were positive for PRRSV by the multiplex cdRT-PCR, and all the positive qRT-PCR samples were also the positive cdRT-PCR samples, indicating a coincidence rate of 94.10% of these two methods (Table 5).

## 4. Discussion

qPCR/qRT-PCR has been widely used in Chinese veterinary laboratories due to its sensitivity, specificity, rapidity, accuracy, and high throughput in detecting viral nucleic acids. qPCR/qRT-PCR provides several advantages over traditional PCR/RT-PCR procedures, including less contamination, rapid performance, and higher sensitivity. The consequence of qPCR/qRT-PCR is determined by the relationship between the threshold cycle (Ct) values and the standard calibration curve [17]. However, the limitations of qPCR/qRT-PCR include the Ct values fluctuate depending on threshold settings, which can be subjective or empirical, and it is less strict in drawing a cutoff line simply using raw Ct values, which can lead to error amplification; the sensitivity to PCR/RT-PCR inhibitors can influence measurement accuracy; and the production of a calibration curve may be a time-consuming and labor-intensive procedure [20]. dPCR/dRT-PCR determines the absolute quantification of nucleic acid without using calibration curves and is less dependent on the reaction efficiency [18,19]. The dPCR/dRT-PCR system depends on limiting dilutions of the PCR/RT-PCR volume and Poisson statistics, and the absolute number of target nucleic acids in the original sample can be calculated by Poisson statistics from the ratio of positive to total partitions [21,22]. Compared with qPCR/qRT-PCR, the main advantages of dPCR/dRT-PCR include absolute quantification independent of calibration curves, increased precision, low susceptibility to PCR/RT-PCR inhibitors, improved accuracy, reliability and reproducibility, and the detection of considerably lower concentrations [20,35]. However, dPCR/dRT-PCR still has some shortcomings, chiefly including the dynamic variation in digital PCR/RT-PCR detection is restricted by the maximum number of partitions; the samples with too high template concentrations need to be properly diluted in advance; the detection flux of some digital PCR/RT-PCR platforms is lower than that of qPCR/qRT-PCR, and the operation steps are more complex; and the use of digital PCR/RT-PCR requires special instruments and consumables, resulting in a greater cost of single detection than that of qPCR/qRT-PCR [20,35,36]. Because there are advantages and disadvantages in both qPCR/qRT-PCR and dPCR/dRT-PCR, they can be selectively used to test viral nucleic acid, depending on the human ability and apparatus of different laboratories. Therefore, a multiplex qRT-PCR and a multiplex cdRT-PCR were developed for the simultaneous detection and differentiation of C-PRRSV, HP-PRRSV, and NL-PRRSV in this study.

PRRSV is one of the most important swine pathogens circulating worldwide. In China, EU-PRRSV and NA-PRRSV have been co-circulating in many pig herds, and currently, NA-PRRSV is the predominant genotype. Furthermore, there are many strains of NA-PRRSV in Chinese pig herds due to the high chance of recombinants and variations [11,37,38,39], and recently, C-PRRSV, HP-PRRSV, and NL-PRRSV are the three major strains of NA-PRRSV that have been identified in the field [9,10,13,14,15,16,27,40]. These viral strains show distinct virulence and cause PRRS with different morbidity rates and mortality rates in different pig herds, and HP-PRRSV has the highest virulence followed by NL-PRRSV and C-PRRSV [7,41,42]. Therefore, it is necessary to identify the circulating strains of PRRSV in order to adopt effective prevention and control strategies. The purpose of this study was to develop a multiplex cdRT-PCR for the differential detection of C-PRRSV, HP-PRRSV, and NL-PRRSV. At the same time, a multiplex qRT-PCR was also developed and used to compare with the multiplex cdRT-PCR in testing the clinical samples. After the optimization of the reaction system and reaction conditions, both methods showed strong specificity, high sensitivity, and good reproducibility. The LOD of the multiplex qRT-PCR and the multiplex cdRT-PCR was 3.20 × 10^0^ copies/μL and 3.20 × 10^−1^ copies/μL (final concentrations), respectively, showing that the LOD of the multiplex cdRT-PCR was 10 times higher than that of the multiplex qRT-PCR. The LOD of the multiplex qRT-PCR in this study was more sensitive [26,29] or similar to [24,28] that of the previous reports, and the LOD of the multiplex cdRT-PCR showed similar situation [31,32]. However, when the template concentration was higher than 3.20 × 10^4^ copies/μL, the dynamic range of the multiplex cdRT-PCR was narrow, and the droplets were completely saturated, which made the Poisson algorithm invalid. These results suggest that the multiplex cdRT-PCR was a more effective method for the accurate quantification of PRRSV compared to the multiplex qRT-PCR, especially when detecting a very low concentration of PRRSV. Finally, 320 clinical samples from Guangxi province from 2020 to 2021 were tested by the two established methods to confirm their application for the detection of clinical samples, and the results show that the positive rate of PRRSV by the multiplex cdRT-PCR was higher than that by the multiplex qRT-PCR.

A total of 320 clinical samples, which were collected from January 2020 to December 2021 in Guangxi province, Southern China, were tested by the developed assays in this study. The results show that the positive rates of C-PRRSV, HP-PRRSV, and NL-PRRSV were 1.88%, 21.56%, and 9.69% by the multiplex qRT-PCR, and 2.19%, 25.31%, and 11.56% by the multiplex cdRT-PCR, respectively. Recently, Liang et al. reported that 50.62% (530/1047) of samples collected from 257 pig farms in Central (Henan province) and South China (Fujian, Guangdong, and Guangxi provinces) in 2016 were positive for PRRSV, and the positive rates of C-PRRSV, HP-PRRSV, and NL-PRRSV were 0.96% (10/1047), 33.52% (351/1047), and 13.94% (146/1047), respectively [43]. Qiu et al. reported that a total of 447 clinical samples from Shandong province, Eastern China, between 2017 and 2018 showed 37.14% (166/447) positive rates of NA-PRRSV, and the positive rates of C-PRRSV, HP-PRRSV, and NL-PRRSV were 1.79% (8/447), 14.32% (64/447), and 16.78% (75/447), respectively, while the co-infection rates of C-PRRSV plus NL-PRRSV and HP-PRRSV plus NL-PRRSV were 0.89% (4/447) and 3.36% (15/447), respectively [27]. Fang et al. reported that 18.82% (1279/6795) of clinical samples from diseased pigs in South China (Guangxi and Guangdong provinces) between 2017 and 2021 were positive for PRRSV, and phylogenetic analysis based on 479 Nsp2 sequences revealed that 85.39% (409/479) were HP-PRRSV and 13.78% (66/479) were NL-PRRSV [16]. Zhou et al. reported that a total of 231 samples from Eastern China (Jiangsu, Zhejiang, Anhui, and Shandong provinces) from 2017 to 2022 showed a 24% (54/231) positive rate of NA-PRRSV, and the phylogenetic analysis of 13 positive samples revealed that 4 samples (30.77%) were C-PRRSV, 5 samples (38.46%) were HP-PRRSV, 2 samples (15.38%) were NL-PRRSV, and 2 samples (15.38%) were the NADC34-like strain [44]. These previous reports, together with our report, suggest that the predominant PRRSV strains in China at present are HP-PRRSV strains, and the proportion of NL-PRRSV strains has increased, while the C-PRRSV strains are still circulating in some pig herds.

PRRSV causes reproductive disorders in sows and respiratory diseases in pigs of all ages. This virus has been persistent in pig herds and still can be detected in lymph nodes until 251 days after infection. The virus can be intermittently shed from the infected pigs, which makes it very difficult to eradicate PRRS once PRRSV has been introduced into pig herds [45,46]. Furthermore, PRRSV causes immunosuppression to the host immune system, including innate and adaptive immunity [47,48,49,50], which aggravates the damage of the disease, and also increases co-infection and secondary infection in pig herds [51,52,53,54]. Vaccination is one of the most effective methods to prevent and control PRRS nowadays. However, PRRS vaccines have only shown full effectivity to homologous strains but have shown limited effectivity to heterogeneous strains [55,56,57,58]. Nowadays, there exists EU-PRRSV, NA-PRRSV, and different strains of NA-PRRSV, such as C-PRRSV, HP-PRRSV, and NL-PRRSV, in some pig herds. The genetic diversity of the circulating strains in the field makes it very hard to select suitable vaccine strains for use in the field. Therefore, it is very important to determine the epidemic strains to ensure that the vaccine strains used are targeted to the clinical epidemic strains. In this study, the developed multiplex qRT-PCR and multiplex cdRT-PCR could detect and differentiate the C-PRRSV, HP-PRRSV, and NL-PRRSV strains in one reaction, which could provide a method to determine the different circulating clinical strains.

## 5. Conclusions

A multiplex qRT-PCR and a multiplex cdRT-PCR, with high sensitivity, specificity, stability, and accuracy, were developed for the simultaneous detection and differentiation of the C-PRRSV, HP-PRRSV, and NL-PRRSV strains of NA-PRRSV. The multiplex qRT-PCR can be used for daily monitoring and detection, as well as a large-scale epidemiological investigation of PRRSV, while cdRT-PCR, besides the abovementioned applications, can be used to accurately quantify the viral loads in samples, especially samples with a very low concentration of the targeted nucleic acids. This is the first report on the multiplex cdRT-PCR to differentiate the C-PRRSV, HP-PRRSV, and NL-PRRSV strains of NA-PRRSV.

## Figures and Tables

**Figure 1 animals-13-00594-f001:**
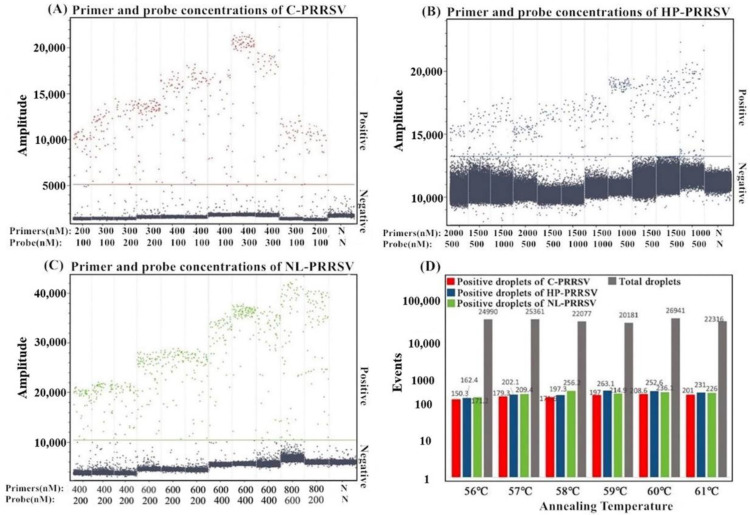
Optimization of the primer and probe concentrations (**A**–**C**) and the annealing temperature (**D**) for the multiplex cdPCR. (**A**–**C**) show the amplification results of p-C-PRRSV, p-HP-PRRSV, and p-NL-PRRSV plasmids (all at final concentrations of 3.20 × 10^2^ copies/μL) with different probe and primer concentrations. N: negative control. (**D**) shows the amplification results of p-C-PRRSV, p-HP-PRRSV, and p-NL-PRRSV plasmids (all at final concentrations of 3.20 × 10^2^ copies/μL) with different annealing temperatures.

**Figure 2 animals-13-00594-f002:**
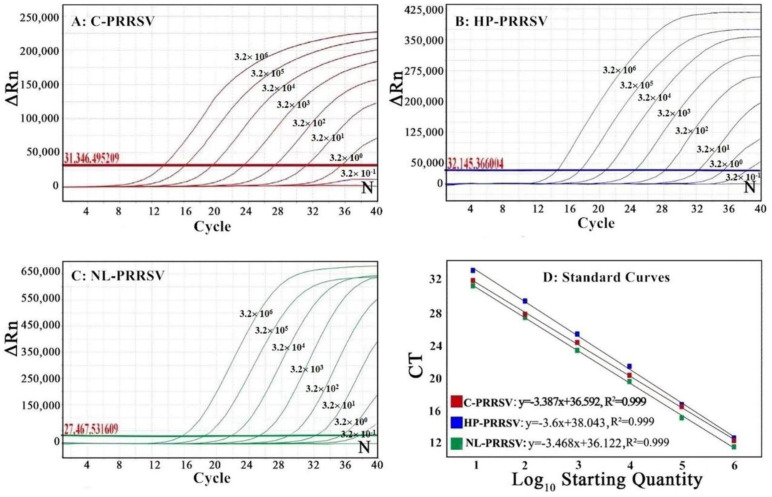
The sensitivity (**A**–**C**) and standard curves (**D**) of the multiplex qRT-PCR. In (**A**–**C**), the final concentrations of the plasmids were from 3.20 × 10^6^ copies/μL to 3.20 × 10^−1^ copies/μL. N: negative control. The red numbers represent the fluorescence values of the threshold baselines.

**Figure 3 animals-13-00594-f003:**
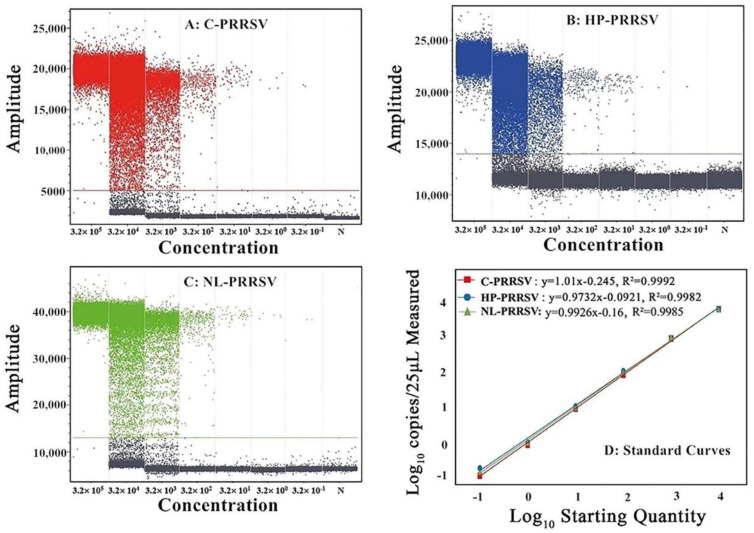
The sensitivity (**A**–**C**) and standard curves (**D**) of the multiplex cdPCR. In (**A**–**C**), the final concentrations of the plasmids were from 3.20 × 10^5^ copies/μL to 3.20 × 10^−1^ copies/μL. N: negative control.

**Figure 4 animals-13-00594-f004:**
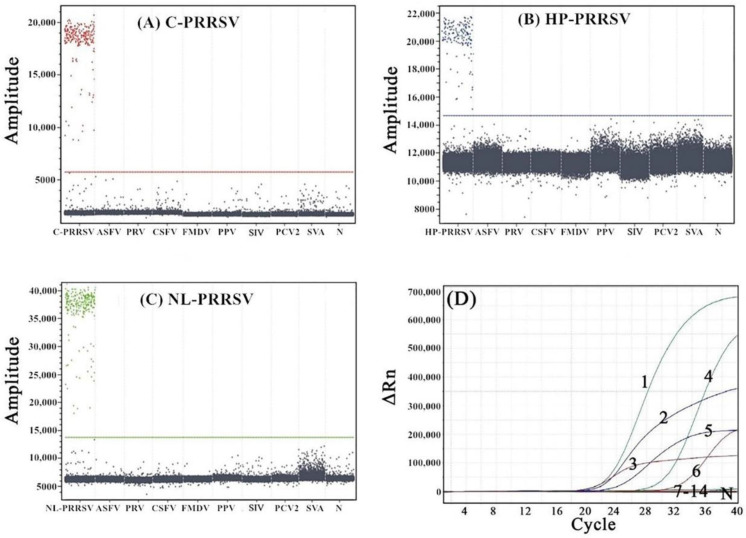
The specificity analysis of the multiplex cdPCR (**A**–**C**) and qRT-PCR (**D**). (**A**–**C**) show the results of the multiplex-cdPCR-specific tests of C-PRRSV, HP-PRRSV, and NL-PRRSV, respectively. (**A**): C-PRRSV, (**B**): HP-PRRSV, and (**C**): NL-PRRSV. (**D**) shows the results of the multiplex-qRT-PCR-specific tests of C-PRRSV, HP-PRRSV, and NL-PRRSV. 1: p-NL-PRRSV; 2: p-HP-PRRSV; 3: p-C-PRRSV; 4: NL-PRRSV; 5: HP-PRRSV; 6: C-PRRSV; and 7–14: ASFV, PRV, CSFV, FMDV, PPV, SIV, PCV2, and SVA, respectively. N: negative control.

**Table 1 animals-13-00594-t001:** Primers and probes for detection of C-PRRSV, HP-PRRSV, and NL-PRRSV.

Primer	Sequence (5′→3′)	Product Size (bp)
C-PRRSV-F	AGTTGGGAAGATTTGGCTGTTA	234
C-PRRSV-R	ACCTGCTGAAACTTACGCCGCG	
C-PRRSV-P	CY5-TCACCGCAATGCATCTTCAGGC-BHQ2	
HP-PRRSV-F	GTCGCGACGTGTCCCCAAGCT	172
HP-PRRSV-R	GCCCATGTTCTGCGATGGT	
HP-PRRSV-P	FAM-CACCAGTTCCTGCACCGCGTAGAACT-BHQ1	
NL-PRRSV-F	AACGTATTGGACACCTCTTTTG	217
NL-PRRSV-R	TGGACCTAATCTTCCTGCGTGGG	
NL-PRRSV-P	VIC-CGGTATTCCAGTCTCGAAAAGC-BHQ1	

**Table 2 animals-13-00594-t002:** The reaction system of the multiplex cdPCR and the multiplex qRT-qPCR.

	Multiplex cdRT-PCR		Multiplex qRT-PCR	
	Volume (μL)	Final Concentration (nM)	Volume (μL)	Final Concentration (nM)
qScript XLT One-Step RT-qPCR ToughMix (2×)	12.5	1×	/	/
Fluorescein sodium salt (1 μM)	2.5	100	/	/
One-Step RT-PCR Buffer (2×)	/	/	12.5	1×
Ex Taq HS (5 μM)	/	/	0.5	100
Primer Script RT Enzyme Mix (5 μM)	/	/	0.5	100
C-PRRSV-F (25 μM)	0.4	400	0.3	300
C-PRRSV-R (25 μM)	0.4	400	0.3	300
C-PRRSV-P (25 μM)	0.3	300	0.2	200
HP-PRRSV-F (25 μM)	1.0	1000	0.4	400
HP-PRRSV-R (25 μM)	1.0	1000	0.4	400
HP-PRRSV-P (25 μM)	0.5	500	0.5	500
NL-PRRSV-F (25 μM)	0.6	600	0.5	500
NL-PRRSV-R (25 μM)	0.6	600	0.5	500
NL-PRRSV-P (25 μM)	0.4	400	0.5	500
Template (total nucleic acids)	2.5	/	2.5	/
RNase-free distilled water	Up to 25	/	Up to 25	/

**Table 3 animals-13-00594-t003:** Repeatability and reproducibility analysis of the multiplex cdRT-PCR and the multiplex qRT-PCR.

Plasmid	Final Concentration (copies/µL)	Intra-Assay for Repeatability						Inter-Assay for Reproducibility					
		Multiplex cdPCR (copies/µL)			Multiplex qRT-PCR(Ct)			Multiplex cdPCR (copies/µL)			Multiplex qRT-PCR (Ct)		
		$\bar{X}$	SD	CV (%)	$\bar{X}$	SD	CV (%)	$\bar{X}$	SD	CV (%)	$\bar{X}$	SD	CV (%)
p-C-PRRSV	3.2 × 10^3^	2334.33	37.45	1.68	24.171	0.05	0.19	2332.55	14.42	0.62	23.81	0.38	1.60
	3.2 × 10^2^	214.7	2.65	1.31	27.71	0.35	1.26	217.30	3.63	1.67	27.67	0.09	0.32
	3.2 × 10^1^	19.10	0.25	1.70	30.89	0.18	0.58	18.90	0.38	2.01	30.67	0.52	1.70
p-HP-PRRSV	3.2 × 10^3^	2324	39.60	1.70	25.76	0.10	0.39	2310	57.20	2.48	25.64	0.10	0.39
	3.2 × 10^2^	196.5	3.90	1.98	29.34	0.21	0.72	201.4	5.30	2.63	29.54	0.19	0.64
	3.2 × 10^1^	19.46	0.45	2.31	32.87	0.28	0.85	19.13	0.13	0.68	32.67	0.22	0.67
p-NL-PRRSV	3.2 × 10^3^	2328.3	37.07	1.59	24.28	0.26	1.07	2308.1	46.42	2.01	24.62	0.39	1.58
	3.2 × 10^2^	250.4	6.70	2.68	27.57	0.16	0.58	252.4	3.30	1.31	27.14	0.14	0.52
	3.2 × 10^1^	23.2	0.55	2.37	30.92	0.12	0.39	22.86	0.57	2.49	30.74	0.24	0.78

**Table 4 animals-13-00594-t004:** Detection results of the clinical samples by the multiplex cdRT-PCR and the multiplex qRT-PCR.

Pathogen	Number	Multiplex qRT-PCR		Multiplex cdRT-PCR		Coincidence Rate (%)	Kappa
		Positive	Positive Rate (%)	Positive	Positive Ratio (%)		
C-PRRSV	320	6	1.88	7	2.19	99.69	0.92
HP-PRRSV	320	69	21.56	81	25.31	96.25	0.90
NL-PRRSV	320	31	9.69	37	11.56	98.13	0.90
C-PRRSV plus HP-PRRSV	320	2	0.63	2	0.63	100	/
C-PRRSV plus NL-PRRSV	320	1	0.31	1	0.31	100	/
HP-PRRSV plus NL-PRRSV	320	15	4.69	20	6.25	98.44	/
C-PRRSV plus HP-PRRSV plus NL-PRRSV	320	1	0.31	1	0.31	100	/

**Table 5 animals-13-00594-t005:** Comparison of the results using the multiplex qRT-PCR and the multiplex cdRT-PCR.

Multiplex qRT-PCR	Multiplex cdRT-PCR			Coincidence Rate	Kappa Value
	Positive	Negative	Total		
Positive	106	0	106	0.941	0.873
Negative	19	195	214		
Total	125	195	320		

## Data Availability

Not applicable.

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
