# Peer review of "Development of a Multiplex Crystal Digital RT-PCR for Differential Detection of Classical, Highly Pathogenic, and NADC30-like Porcine Reproductive and Respiratory Syndrome Virus"

_animals, 2023, doi:10.3390/ani13040594_

Round 1

Author Response

Response to Reviewer’s Comments

Dear editor,

We are very grateful for the reviews provided by the editors and each of the external reviewers of this manuscript. Coauthors very much appreciated the constructive comments on this manuscript. The comments have been useful in improving the manuscript. We have taken them fully into account in revision. We are submitting the corrected manuscript with the suggestion incorporated the manuscript. The manuscript has been revised as per the comments given by the reviewer (with tracked changes), and our responses to all the comments are as follows:

Reviewer #1:

This manuscript developed both a multiplex qRT-PCR and a multiplex cdRT-PCR test for C-PRRSV, HP-PRRSV, NL-PRRSV strains of NA-PRRSV. The authors explored conditions to optimize the amplification reaction and did sensitivity, specificity, and repeatability analysis. Specifically, this work also tested 320 clinical samples.

Several points need to be addressed in a revision prior to publication for clarity as detailed below.

Comment 1: In Sensitivity analysis, the authors reported the LoD of their tests but the definition and calculation method of LoD need to be described (Page 9 Line 287).

Response: The definition and calculation method of LOD was described in the revised manuscript. Please see Lines 339-342 of the revised manuscript.

Comment 2: How does the LoD of this method compared to the other previously reported PCR for PRRS virus?

Response: The LOD of this method was compared with the other previously reported PCR for PRRS virus in the Discussion section in the revised manuscript. Please see Lines 449-451 of the revised manuscript.

Comment 3: The results of the total 320 clinical samples showed that the positive rates of C-PRRSV, HP-PRRSV, and NL-PRRSV were 1.88%, 21.56%, and 9.69% by the multiplex qRT-PCR, and 2.19%, 25.31%, and 11.56% by the multiplex cdRT-PCR, respectively. What are the potential reasons for the different positive rates of the same batch of samples? What are the false positive rate and false negative rate of both tests?

Response: The positive rates of the multiplex cdRT-PCR were higher than those of the multiplex qRT-PCR due to the higher sensitivity of the former method. The coincidence rate of the results by two methods was calculated and showed in Table 5 in the revised manuscript. Please see Page 12 Table 5 in the revised manuscript.

Comment 4: Page 12 Line 348 “The dPCR/dRT-PCR determines the absolute quantification of nucleic acid without using calibration curves and without relying on the reaction efficiency”. Why does dPCR does not rely on the reaction efficiency?

Response: The original statement is not accurate. In the revised manuscript, “without relying on the reaction efficiency” was changed to “is less dependent on the reaction efficiency”. Please see Lines 412-413 in the revised manuscript.

Comment 5: In Figure 2 and 3, number 1-8 or 1 to 7 are better to be changed to the real concentrations for clarification.

Response: The real concentrations were labled in Figure 2 and Figure 3 according to the reviewer’s suggestion. Please see Figure2 and Figure 3 in the revised manuscript.

Comment 6: What the red numbers represent in Figure 2 should be described either in caption or in context.

Response: The red numbers represent in Figure 2 have been described in caption according to the reviewer’s suggestion. Please see Lines 298-299 in the revised manuscript.

Reviewer 2 Report

Authors develop a multiplex real-time quantitative RT-PCR (qRT-PCR) and a multiplex Crystal digital RT-PCR (cdRT-PCR) for differential detection of classical PRRSV (C-PRRSV), highly pathogenic PRRSV (HP-PRRSV), and NADC30-like PRRSV (NL-PRRSV). The specificity, sensitivity, and repeatability of qRT-PCR and cdRT-PCR are examined completely and carefully, and 320 clinical samples examination show a good coincidence. The article has the potential to be accepted, but a few points have to be clarified.

1. L251, the Figure legend is incomplete. The sentence in line 253 should continue close after the sentence in line 251 - 252. There are only 6 annealing temperatures that should be labeled.

2. In Figure 2, the number 1 to 7 in figure A, B, and C are meaning the concentration of concentrations of plasmids should label clearly.

3. In Figure 3, the X-axis is meaning the concentration of plasmids, the X-axis level 1 to 7 are not suitable.

4. Line 288, the concentration of plasmid templates used from 3.2 ×10^6 copies/μL to 3.20 ×10^0 copies/μL to determine the LOD of the cdRT-PCR. The results showed that the LOD of plasmids was 3.20 ×10^-1 copies/μL (final concentrations). Please explain why the LOD concentration was lower than the lowest concentration of the template tested.

5. Line 322, the examination of the clinical sample showed that multiplex cdRT-PCR was 10 times higher sensitivity than that of the multiplex qRT-PCR and multiplex cdRT-PCR detected more positive clinical samples than the multiple qRT-PCR. Are the positive qRT-PCR samples all included in the positive cdRT-PCR samples?

Author Response

Response to Reviewer’s Comments

Dear editor,

We are very grateful for the reviews provided by the editors and each of the external reviewers of this manuscript. Coauthors very much appreciated the constructive comments on this manuscript. The comments have been useful in improving the manuscript. We have taken them fully into account in revision. We are submitting the corrected manuscript with the suggestion incorporated the manuscript. The manuscript has been revised as per the comments given by the reviewer (with tracked changes), and our responses to all the comments are as follows:

Reviewer #2:

Authors develop a multiplex real-time quantitative RT-PCR (qRT-PCR) and a multiplex Crystal digital RT-PCR (cdRT-PCR) for differential detection of classical PRRSV (C-PRRSV), highly pathogenic PRRSV (HP-PRRSV), and NADC30-like PRRSV (NL-PRRSV). The specificity, sensitivity, and repeatability of qRT-PCR and cdRT-PCR are examined completely and carefully, and 320 clinical samples examination show a good coincidence. The article has the potential to be accepted, but a few points have to be clarified.

Comment 1: L251, the Figure legend is incomplete. The sentence in line 253 should continue close after the sentence in line 251 - 252. There are only 6 annealing temperatures that should be labeled.

Response: Done. Please see Lines 251-256 in the revised manuscript.

Comment 2: In Figure 2, the number 1 to 7 in figure A, B, and C are meaning the concentrations of plasmids should label clearly.

Response: The concentrations of plasmids have been labeled in Figure 2. Please see Page 8 Figure 2 in the revised manuscript.

Comment 3: In Figure 3, the X-axis is meaning the concentration of plasmids, the X-axis level 1 to 7 are not suitable.

Response: The concentrations of plasmids have been labeled in Figure 3. Please see Page 9 Figure 3 in the revised manuscript.

Comment 4: Line 288, the concentration of plasmid templates used from 3.2 ×10^6 copies/μL to 3.20 ×10^0 copies/μL to determine the LOD of the cdRT-PCR. The results showed that the LOD of plasmids was 3.20 ×10^-1 copies/μL (final concentrations). Please explain why the LOD concentration was lower than the lowest concentration of the template tested.

Response: The 25 μL reaction system contained 2.5 μL template, therefore the final concentrations of the templates were 10-fold dilution of the initial concentrations in the reaction system. In this study, the initial concentrations of plasmid templates from 3.2 ×106 copies/μL to 3.20 ×100 copies/μL were used to determine the LOD of the cdRT-PCR, and the final concentrations ranged from 3.2 ×105 copies/μL to 3.20 ×10-1 copies/μL. In this study, the LOD of each template was 3.20 ×100 copies/μL of initial concentration, so the final concentration was 3.20 ×10-1 copies/μL. The reason was explained in the Results section. Please see Lines 339-342 and Lines 348-350 in the revised manuscript.

Comment 5: Line 322, the examination of the clinical sample showed that multiplex cdRT-PCR was 10 times higher sensitivity than that of the multiplex qRT-PCR and multiplex cdRT-PCR detected more positive clinical samples than the multiple qRT-PCR. Are the positive qRT-PCR samples all included in the positive cdRT-PCR samples?

Response: The positive qRT-PCR samples are all included in the positive cdRT-PCR samples. The coincidence rate of the results by two methods was calculated and showed in Table 5 in the revised manuscript. Please see Page 12 Table 5 in the revised manuscript.
